# Exploring Maternal Socio-Demographic Factors Shaping Children’s Dietary Patterns in Brazil: Results from the 2019 National Health Survey

**DOI:** 10.3390/ijerph21080992

**Published:** 2024-07-29

**Authors:** Flávia dos Santos Barbosa Brito, Emanuela Santos da Costa, Ariane Cristina Thoaldo Romeiro, Debora Martins dos Santos, Alexandre dos Santos Brito, Alessandra Silva Dias de Oliveira, Amanda Rodrigues Amorim Adegboye

**Affiliations:** 1Nutrition Institute, Rio de Janeiro State University, São Francisco Xavier Street, 524, Rio de Janeiro 20550-900, Brazil; emanuelaccosta@gmail.com (E.S.d.C.); deborams@gmail.com (D.M.d.S.); alessanutri@hotmail.com (A.S.D.d.O.); 2Centro Universitário Serra dos Órgãos, Teresópolis 25964-004, Brazil; 3Instituto de Estudos em Saúde Coletiva, Universidade Federal do Rio de Janeiro, Rio de Janeiro 21941-598, Brazil; britoalexandre@uol.com.br; 4Centre for Agroecology, Water and Resilience (CAWR), Coventry University, Coventry CV8 3LG, UK; 5Centre for Healthcare Research, Coventry University, Coventry CV1 5FB, UK

**Keywords:** child nutrition, complementary feeding, dietary pattern, socio-demographic factors, mothers

## Abstract

This study aimed to identify the dietary patterns of Brazilian children aged 6–23 months and to investigate their association with maternal socio-demographic factors. Data from the 2019 Brazilian National Health Survey were used in this cross-sectional study. Mothers of 1616 children aged 6–23 months reported on their children’s dietary intake. Dietary patterns were identified using principal component analysis, and their associations with maternal socio-demographic characteristics were assessed using linear regression models. The first consisted of healthy patterns and the second, unhealthy ones. Linear regression showed that adherence to a healthy dietary pattern was higher among children of mothers who were older (β = 0.02, *p* = 0.01), had more years of education (β = 0.49, *p* = 0.04), reported living with a partner (β = 0.29, *p* = 0.01), and resided in an urban area (β = 0.35, *p* = 0.01). Conversely, adherence to the unhealthy pattern was positively associated with mothers who declared themselves as black or brown (β = 0.25, *p* = 0.03). Our results show that older mothers with higher levels of education and paid work and who live with a partner are more likely to contribute to their children’s healthy eating patterns. We conclude that socio-demographic factors may influence the quality of the food offered to children. Nevertheless, advocating for public policies promoting nutritious complementary diets emphasising fresh and minimally processed foods remains crucial for children whose mothers do not possess these favourable socio-demographic characteristics.

## 1. Introduction

The first two years of a child’s life are a critical period for optimal growth, health, and behavioural development [1]. It is recommended to begin complementary feeding at six months and continue this until 23 months, although breastfeeding may continue beyond this period. Complementary feeding provides essential nutrients when breast milk or formula milk alone is insufficient to meet nutritional requirements [2,3]. The types of foods offered and how they are introduced to children can shape their eating preferences and attitudes towards food [3]. Children with healthy diets are more likely to become adults who are able to make healthier food choices [3,4]. On the other hand, children with unhealthy diets have a higher risk of developing chronic non-communicable diseases, such as obesity, dyslipidaemia, type 2 diabetes, and hypertension [5,6,7].

In recent years, there has been a significant shift in the dietary patterns of Brazilians. The consumption of fresh or minimally processed foods and culinary ingredients has decreased, while the consumption of ultra-processed foods (UPFs) has increased [8,9,10]. According to Louzada et al., UPFs accounted for almost 20% of the total energy consumed by the Brazilian population in 2017–2018 [10]. This increase in consumption is not limited to Brazil alone but is also observed worldwide. National surveys indicate that ultra-processed foods account for at least half of the total energy intake in some high-income countries, such as the United Kingdom. In middle-income countries, these foods contribute between one-fifth and one-third of the total energy consumed [11,12,13,14].

Studies revealed that the consumption of UPF is often influenced by social and family environment [2,15,16]. Mothers play an important role in shaping their children’s dietary patterns, especially during the first two years of life. They are responsible for food choice, availability, and access, and for the preparation of food in the household [17,18]. The environment that mothers create for their children can either promote healthy dietary patterns or lead to adverse effects on their nutrition, leading to unhealthy dietary behaviours that can increase the risk of excessive weight gain and chronic non-communicable diseases [2,18,19,20].

Although the importance of nutrition for development in the first two years of life is well documented [1,2,3], it is important to understand how maternal socio-demographic factors affect children’s dietary patterns in the early years of life. We believe that these maternal characteristics influence the quality of the food that is offered to children. This knowledge can inform the development of guidelines or targeted interventions to promote healthy eating habits during childhood. However, few studies have investigated how specific Brazilian maternal socio-demographic factors influence children’s dietary patterns [16,17,18,19,20,21]. This study aims to address this gap by identifying children’s dietary patterns and assessing their association with various maternal socioeconomic factors.

## 2. Materials and Methods

### 2.1. Design and Study Population

This is a cross-sectional, nationally representative household-based study using secondary data from the 2019 edition of the Brazilian National Health Survey (PNS 2019). The survey was conducted by the Brazilian Institute of Geography and Statistics (IBGE) in collaboration with the Ministry of Health and educational and research institutions. Data were collected by trained interviewers between August 2019 and March 2020 [9].

The PNS 2019 is a population-based survey that is representative of the population resident in private households in Brazilian territory. It used a complex sample design with three stages. In the first stage, primary sampling units (PSUs), represented by census tracts, were randomly selected from a master file of census tracts stratified by geographical region and urban/rural situation. In the second stage, households were selected from the PSUs. In the third stage, adult respondents were randomly selected from each household for in-depth interviews [22,23]. The final sample comprised 94,114 households that underwent an interview, yielding a response rate of 93.6% [23].

Sampling weights were defined for the primary sampling units, households, and all their residents. Further details on the sampling procedure, weighting factors, and data collection can be found in previous publications of PNS 2019 [23].

The data collection of the PNS 2019 was based on a questionnaire divided into three sections with different modules (A to Z) to investigate the characteristics of the household and its residents in addition to the collection of individual data on the eligible residents. Module L of this questionnaire included questions on children under two years of age. It is important to note that the youngest child was selected if there was more than one child under two years of age in the household [8,22,23]. The questionnaire can be found in the Appendix A.

In this study, we used data from women who reported having given birth between 28 July 2017 and 27 July 2019 and who answered module L questions in the survey about their respective children who were aged 6–23 months. This resulted in 1616 mother–child pairs.

### 2.2. Study Variables

#### 2.2.1. Dietary Intake

The children’s dietary intake was evaluated through closed-ended questions regarding their consumption of specific foods within the past 24 h. Mothers were asked yes-or-no questions about whether their children had consumed various food groups from the morning prior to the data collection until the morning of the assessment. These food groups included non-breast milk or dairy products, fruits or natural juices, vegetables, beans or legumes, meat or eggs, potatoes and tubers, cereals and their derivatives, artificial juices, cookies or cake, sweets, and soft drinks. The frequency of consumption for each food item was then calculated based on the responses provided.

#### 2.2.2. Maternal Socio-Demographic Factors

The following maternal socio-demographic variables were included in the survey: age (years), education (complete elementary school: no education, incomplete elementary school, and complete elementary school; complete secondary school: incomplete secondary school; complete secondary school and incomplete higher education; complete higher education), ethnicity (white; black or brown; east Asian or Indigenous), living with a partner (no/yes), area of residence (urban or rural), paid work (no/yes), and per capita household income, in minimum wages (MW). Variable paid work was defined as any work for which money was paid in the last seven days. Mothers who performed unpaid or voluntary work and those who lived only on income or help from relatives and friends were included in the category of no paid work. Per capita household income was estimated by dividing the total family income by the number of household residents and expressed in minimum wages. In 2019, the minimum wage in Brazil was BRL 998.00.

#### 2.2.3. Other Variables

The questionnaire also included data on maternal lifestyle. Mothers reported whether they had participated in any type of physical activity or sport in the last three months (no/yes) and in smoking: non-smoker, former smoker, and smoker (no/yes). Individuals who answered “former smoker” and “smoker” were considered smokers, regardless of the number of cigarettes, frequency, and/or duration of the smoking. Information about the age of the child in months was extracted from the children’s questionnaire and used as an adjustment variable in the analysis.

### 2.3. Statistical Analysis

All analyses were performed using the Stata statistical package, version 16.0 (StataCorp LP, College Station, TX, USA), accounting for complex survey design and sampling weights employed using the “*svy*” command. Categorical variables were presented as relative frequencies (%), while continuous variables were represented as means for continuous variables and their respective 95% confidence intervals (95%CI) for continuous variables. Subsequently, a descriptive analysis of the relative frequency (%) of consumption related to each food group encompassed within the questionnaire was conducted.

The dietary patterns were obtained by factor analysis extraction using principal component analysis (PCA). The Bartlett test and the Kaiser–Mayer–Olkin (KMO) coefficient were used to verify the applicability of the method [24]. Using Kaiser’s classification for KMO values [25], the following ranges were applied to characterise the levels of acceptance: 0.00 to 0.49 as unacceptable, 0.50 to 0.59 as poor, 0.60 to 0.69 as fair, 0.70 to 0.79 as moderate, 0.80 to 0.89 as meritorious, and 0.90 to 1.00 as a perfect fit. For the Bartlett test, we assumed a type I error rate of 5%. To determine the number of factors, we considered (i) eigenvalues > 1 and (ii) the inflection point of the eigenvalues from the Cattell scree test (scree plot). Varimax orthogonal rotation was used to facilitate the interpretation of the factors. Factor loadings greater than 0.30 were considered to represent dietary patterns. The total variance explained by each factor was also considered to determine the number of factors to be retained. Cattell’s scree plots and eigenvalues >1 were also considered for pattern selection. The orthogonal varimax transformation was used to facilitate the interpretation of the factor results. The naming of the identified patterns was based on interpretability and the characteristics of the food groups retained in each pattern. Factor scores were predicted for each child in the study. These scores were calculated by adding the standardised values in each pattern’s z-score for the food groups. A higher score indicates greater adherence to the dietary pattern.

Finally, associations between the children’s factor scores for each dietary pattern identified (dependent variables) and the maternal socio-demographic factors (independent variables) were examined using crude and multiple linear regression models, with associations described using β-coefficients, 95% confidence intervals (95% CI), and *p*-values (*p*). Maternal socio-demographic factors that had a *p* < 0.20 in the crude regression model were included in the multiple linear regression model using a backward stepwise procedure, retaining only those that were statistically significant (*p* < 0.05).

The multicollinearity across independent variables was calculated using a variance inflation factor (VIF) for the estimated linear regression coefficients. Following the criteria established by Johnston et al., VIFs of 2.5 or greater were considered indicative of substantial multicollinearity [26]. The analysis revealed no presence of multicollinearity within the model. Furthermore, the normality of the residuals was confirmed through residual distribution analysis, suggesting an appropriate fit of the model. 

### 2.4. Data Availability

The datasets analysed are freely available on the website of the Brazilian Institute of Geography and Statistics (IBGE) at the following URL: https://www.ibge.gov.br/estatisticas/sociais/saude/29540-2013-pesquisa-nacional-de-saude.html?edicao=9177&t=microdados (accessed on 22 July 2024).

### 2.5. Ethical Statement

The 2019 edition of the National Health Survey project was submitted to the National Research Ethics Committee/National Health Council and approved in accordance with No. 3.529.376. issued on 23 August 2019. The study was also approved by the Coventry University Ethics Committee (P134794).

## 3. Results

Table 1 shows the socio-demographic factors of 1616 mothers of children aged 6–23 months. Most mothers identified themselves as black or brown (66.3%), had completed secondary education (42%), and lived in urban areas (82.4%). In addition, 62.1% of the mothers were not employed, and 79.2% lived with a partner. The mean age of the mothers was 29 years old and ranged from 28.4 to 29.6 years. 

The majority of children consumed unprocessed and minimally processed foods: fruits or natural juices (84.1%), beans or other legumes (80.1%), and vegetables (77.4%) on the previous day. However, they also consumed UPF; 67.6% of children consumed cookies or cake and 30.8% sweets, candies, or sugary foods. In addition, our results indicate that 74.2% of children consumed at least one UPF on the previous day (Table 2).

Table 3 presents the assessment of adequacy for the dietary factor analysis, revealing a moderate KMO value of 0.80, with Bartlett results showing a *p*-value < 0.01, indicating adequacy for factor analysis. The PCA identified two dietary patterns and explained 42% of the total variance in the dietary intake (Table 3). The first pattern, labelled “healthy”, consisted of fruits or natural juices, vegetables, beans or other legumes, meat or eggs, potatoes and other tubers and roots, and cereals and derivatives and was the most representative of the dietary intake of this population, accounting for 22.2% of the total variance. The second pattern, “unhealthy,” included UPFs such as artificial juices, cookies or cake, sweets or other foods with sugar, and soft drinks, and accounted for 19.8% of the total variance (Table 3).

Table 4 shows the crude and adjusted associations between maternal socio-demographic factors and children’s dietary patterns. In the adjusted linear regression model, adherence to the healthy dietary pattern was higher among children of mothers who were older, had more years of education, reported living with a partner, and resided in an urban area. On the other hand, adherence to the unhealthy pattern was higher among children of mothers who identified themselves as black or brown.

## 4. Discussion

This study aimed to identify the dietary patterns of Brazilian children aged 6–23 months and to investigate their association with maternal socio-demographic factors. We hypothesised that distinct dietary patterns would emerge and that these patterns would be associated with specific socio-demographic characteristics of the mothers.

The present study identified two dietary patterns—healthy and unhealthy. The healthy pattern explained the highest proportion of total variance and best represented the dietary intake of Brazilian children under two years of age. This pattern included unprocessed or minimally processed food groups related to fruits, beans or legumes, vegetables, meat or eggs, and non-breast milk. These results suggest that Brazilian children tend to have a healthy dietary pattern that is in line with the dietary guidelines for Brazilian children under 2 years of age [3]. These guidelines recommend the introduction of an appropriate and healthy complementary diet, primarily comprising fresh or minimally processed foods, starting at 6 months of age and alongside continued breastfeeding. [3]. The second pattern, labelled “unhealthy”, was characterised by the consumption of UPF. Evidence from representative studies of 11 countries between 2001 and 2015 shows that the greater inclusion of UPFs in the diet is associated with worse diet quality [27]. These included artificially sweetened drinks, sweets, cookies, and soft drinks. Although most children in the present study followed the healthy pattern, 74.2% of children consumed at least one UPF on the previous day. In line with these findings, data from the first Brazilian National Survey on Child Nutrition (ENANI-2019) revealed a concerning prevalence of UPF consumption among children aged 6–23 months, reaching 80.5% [27]. Cainelli et al. found that 79.4% of children aged between 1 and 2 years consumed some type of UPF and that the consumption of such foods was associated with household socioeconomic and demographic factors [28]. This suggests that mothers with a more stable socioeconomic status are more likely to promote healthier eating habits among their children [28]. These findings are consistent with international and national studies that have reported similar associations [28,29,30].

These high UPF consumption rates are particularly alarming, given the trend of their early introduction into the complementary feeding of children under two years of age, not only in Brazil but globally. This early exposure to UPFs has been associated with an increased risk of developing dyslipidaemia, overweight, obesity, and other cardiovascular diseases [6,13,29,31,32,33,34,35,36]. 

Our results showed a positive association between favourable maternal socio-demographic characteristics and healthy dietary patterns in children on the previous day. Children who had a healthy dietary pattern were more likely to have older mothers with higher levels of education, employment, and who lived with a partner. This suggests that mothers with a more stable socioeconomic status are more likely to promote healthier eating habits among their children. These findings are consistent with both international and national studies that have reported similar associations [6,30,37,38,39,40,41]. For example, Campos et al. highlighted that mothers can positively or negatively influence their children’s eating habits and diet quality [42]. In another study, Paramashanti et al. highlight that mothers or caregivers with higher levels of education are likely to be more knowledgeable about child nutrition and health literacy and to better understand the messages conveyed about child nutrition [43]. Similarly, a systematic review by Gutiérrez-Camacho et al. provides evidence of a positive association between maternal education level, higher household income, and higher maternal age and healthy dietary patterns, and a negative association between these factors and unhealthy dietary patterns [44]. These factors are important in providing access to healthy foods, better quality products, and adequate information on healthy dietary patterns, as highlighted in the dietary guidelines for Brazilian Children under 2 Years of Age [3].

In this study, having a working mother was positively associated with a healthy dietary pattern in children. Controversially, other studies have found that children and adolescents of working mothers consume more UPFs [8,31,45,46,47,48]. Cawley and Liu found that working mothers allocated significantly less time to activities such as grocery shopping, cooking, eating, and engaging in playing with their children. Additionally, they were more likely to buy ready-made foods [49]. Several factors could contribute to this disparity. Firstly, the definition of a “healthy dietary pattern” may vary between studies, leading to differences in interpretation. Secondly, cultural or socioeconomic factors specific to the studied populations may influence dietary behaviours differently. Additionally, individual family dynamics and support systems could play a significant role. While working mothers may have less time for certain activities like grocery shopping and cooking, they might compensate through alternative strategies, such as meal planning, seeking out healthier convenience options, involving other family members in meal preparation, or paying for housekeeping help.

Our findings regarding ethnicity align with the existing literature. Pereira et al. observed a higher likelihood of children consuming UPFs if they were born to a black mother [50]. Similarly, Lacerda et al. found that children aged 6–23 months with white mothers or caregivers had approximately three times the prevalence of adequate nutrition, as measured by minimum dietary diversity without ultra-processed foods, compared with children with black mothers or caregivers [27]. Specifically, the prevalence was 11.2% (95%CI: 7.3–15.1) among children with white mothers or caregivers versus 3.6% (95%CI: 1.7–5.6) among those with black mothers or caregivers [27].

Silva et al. explored food insecurity from an intersectional perspective, highlighting that advancements in education and income in Brazil have not been uniformly distributed, as the black population, especially women, experience social disadvantages [51]. The authors emphasised that it is still necessary to pay attention to the specific dynamics of this group and, in this sense, there is a need to incorporate intersectionality in the development of public policies to combat hunger.

### Limitations

The study has some limitations. Firstly, its cross-sectional design prevents causal relationships from being established. Thus, inherent in the study’s design, we analysed associations rather than causality. The reliance on information about food consumption pertaining only to the day preceding the survey may not accurately represent children’s usual consumption. As with all surveys that are based on self-reporting, there is a potential for recall and social desirability bias. However, the use of a large and nationally representative sample with PNS data may have minimised the effect of variability in food consumption [52,53]. Despite its limitations, this study has important strengths. Our results strengthen the current scientific literature by finding two dietary patterns in Brazilian children that are discussed in the dietary guidelines for Brazilian children under 2 years of age. In addition, we have shown that the maternal socio-demographic factors to which the children are exposed may influence these patterns. In this study, we observed a higher likelihood of mothers identifying as black or brown to adhere to the unhealthy dietary pattern, even after controlling for other socio-demographic factors. The discussion of ethnicity in health and nutrition studies is a step forward. However, these associations might have been affected by residual confounding, particularly regarding socioeconomic status variables. Therefore, more studies are still needed to understand ethnic/racial dynamics and their impact on intergenerational health issues.

By using PNS data, the present study delved into the exploration of how specific maternal socio-demographic factors intertwine with children’s dietary patterns. Understanding these dynamics could serve as a cornerstone for formulating effective strategies aimed at nurturing good nutrition and fostering healthier dietary behaviours among children in Brazil and, potentially, in other South American countries.

Alongside promoting public policies fostering healthy diets rooted on fresh or minimally processed foods among mothers [3], emphasis should be placed on disseminating educational resources, particularly targeting younger mothers with lower education levels.

## 5. Conclusions

Our results show that older mothers with higher levels of education, paid work, and living with a partner are more likely to contribute to their children’s healthy eating patterns, suggesting that these socio-demographic factors may be more likely to influence healthier eating habits. Accordingly, public policies promoting nutritious complementary diets that emphasise fresh or minimally processed foods should be offered to all mothers from different socioeconomic levels, with the goal of establishing long-term healthy habits and creating pleasant eating patterns in their children while becoming aware of behavioural determinants that favour malnutrition and eating disorders. In addition, monitoring infant feeding practices through population-based surveys is essential to assess changes in nutritional indicators and to explore potential determinants and consequences of dietary choices in this population.

## Figures and Tables

**Table 1 ijerph-21-00992-t001:** Maternal socio-demographic factors of children aged 6–23 months. Data from the Brazilian National Health Survey, 2019, Brazil (n = 1616).

		Mean	95%CI	
Mother’s age (Years)		28.9	28.4	29.6
Child’s age (Months)		14.6	14.0	15.1
		**Percentage (%)**	**95%CI**	
Education				
	Complete elementary school	39.8	35.2	44.5
	Complete secondary school	42.0	37.7	46.4
	Complete higher education	18.2	14.7	22.45
Ethnicity				
	White	32.7	31.8	34.5
	Black or Brown	66.3	64.5	68.8
	East Asian or Indigenous	1.1	0.6	1.9
Paid work				
	No	37.9	34.3	41.6
	Yes	62.1	58.4	65.7
Living with a partner				
	No	20.3	16.6	24.7
	Yes	79.2	75.3	83.4
Physical activity				
	No	74.3	70.1	78.1
	Yes	25.7	21.9	29.9
Smoking				
	No	92.4	89.9	94.3
	Yes	7.6	5.7	10.1
Area of residence				
	Urban	82.4	80.6	84.1
	Rural	17.6	15.9	19.5
Per capita household income, in minimum wages (MW)		0.93	0.78	1.09

95%CI: 95% confidence interval; Complete elementary school: no education, incomplete elementary school, and complete elementary school; complete secondary school: incomplete secondary school; complete secondary school and incomplete higher education; complete higher education). Minimum wage (MW) value in 2019: BRL 998.00.

**Table 2 ijerph-21-00992-t002:** Frequency (%) of consumption of foods on the previous day among children aged 6–23 months. Data from the Brazilian National Health Survey, 2019, Brazil (n = 1616).

	%	95%CI	
Unprocessed or minimally processed foods			
Fruits or natural juices	84.1	80.6	87.1
Beans or other legumes	80.1	78.7	81.4
Vegetables	77.4	73.7	80.7
Meat or eggs	77.3	75.7	78.7
Cereals and derivatives	75.2	71.3	78.1
Non-breast milk or dairy products	74.1	70.5	77.5
Potatoes and other tubers and roots	62.7	58.2	67.0
Ultra-processed foods (UPFs)			
Cookies or cake	67.6	63.5	71.4
Sweets, candies, or other sugary foods	30.8	25.8	34.5
Artificial juices	23.1	19.1	27.7
Soft drinks	11.9	9.60	14.8
Number of types of UPFs consumed			
0	25.8	22.6	29.4
1	38.1	34.3	42.0
2	18.7	15.7	22.3
3	12.5	9.3	16.4
4	4.9	3.5	6.9

95%CI: 95% confidence interval.

**Table 3 ijerph-21-00992-t003:** Rotated factor loadings for the two dietary patterns identified by factor analysis (principal component) of children aged 6–23 months. Data from the Brazilian National Health Survey (PNS), 2019, Brazil (n = 1616).

Consumed Food	Dietary Pattern		h^2 a^
	Healthy	Unhealthy	
Non-breast milk or dairy products			0.06
Fruits or natural juices	0.41		0.38
Vegetables	0.52		0.62
Beans or other legumes	0.33		0.43
Meat or eggs	0.40		0.55
Potato and other tubers and roots	0.41		0.39
Cereals and derivatives	0.30		0.42
Artificial juices		0.45	0.40
Cookies or cake		0.35	0.39
Sweets, candies, or other sugary foods		0.51	0.54
Soft drinks		0.46	0.43
Eigenvalues	2.96	1.66	
Proportional variance (%)	22.17	19.81	

The items indicated in bold showed a factor loading ≥0.3 or ≤−0.3; ^a^ Commonality: proportion of the variance of each variable explained by the extracted factors. Accumulated variance (%): 41.9; Kaiser–Meyer–Olkin (KMO) value: 0.8; Bartlett’s sphericity test: *p*-value < 0.01.

**Table 4 ijerph-21-00992-t004:** Crude and multiple linear regression of dietary patterns with maternal socio-demographic factors among children aged 6–23 months. Data from the Brazilian National Health Survey (PNS), 2019, Brazil (n = 1616).

		Healthy								Unhealthy							
		Crude β	95%CI		*p*	Multiple ^a^ β	95%CI		*p*	Crude β	95%CI		*p*	Multiple ^a^ β	95%CI		*p*
Age (Years)		0.39	0.02	0.06	0.01	0.02	0.01	0.03	0.01	−0.02	−0.03	−0.01	0.01	−	−	−	−
Education																	
	Complete elementary	Ref.				Ref.				Ref.				Ref.			
	Complete secondary school	0.37	0.39	0.71	0.03	0.38	0.21	0.55	0.01	−0.29	−0.56	0.02	0.10	−0.24	−0.50	0.02	0.12
	Complete higher education	0.97	0.66	1.28	0.01	0.49	0.20	0.78	0.01	−0.67	−0.99	−0.36	0.01	−0.52	−0.85	−0.20	0.01
Ethnicity																	
	White	Ref.				Ref.				Ref.							
	Black or Brown	−0.27	−0.56	−0.07	0.01	−0.18	−0.36	−0.04	0.04	0.46	0.20	0.73	0.01	0.25	0.04	0.48	0.03
	East Asian or Indigenous	−0.49	−1.48	0.50	0.33	−0.44	−1.02	0.15	0.14	−0.16	−1.20	0.88	0.76	−0.36	−1.05	0.33	0.31
Paid work																	
	No	Ref.				−				Ref.				−			
	Yes	0.44	0.16	0.73	0.01	−	−	−	−	0.11	−0.36	0.25	0.14	−	−	−	-
Living with a partner																	
	No	Ref.				Ref.				Ref.				−			
	Yes	0.37	−0.07	0.81	0.01	0.29	0.11	0.47	0.01	−0.05	−0.21	0.11	0.57	−	−	−	-
Physical activity																	
	No	Ref.				−				Ref.				−			
	Yes	0.53	0.25	0.82	0.01	−	−	−	−	−0.01	−0.27	0.25	0.93	−	−	−	-
Smoking																	
	No	Ref.								Ref.							
	Yes	−0.35	−0.75	0.06	0.09	−				0.37	−0.01	0.74	0.06	−	−	−	-
Area of residence							−	−	−	−							
	Rural	Ref.				Ref.				Ref.				−			
	Urban	0.45	0.14	0.76	0.01	0.35	0.17	0.52	0.01	0.06	−0.24	0.36	0.67	−	−	−	-
Per capita household income, in minimum wages (MW)		0.13	−0.02	0.28	0.08	−	−	−	−	−0.12	−0.18	−0.07	0.01	−	−	−	−

^a^ The model included the child’s age (in months) as a covariate. 95% CI: 95% confidence interval. Significant *p*-values: *p* < 0.05. Complete elementary school: no education, incomplete elementary school, and complete elementary school; complete secondary school: incomplete secondary school; complete secondary school and incomplete higher education; complete higher education). Minimum wage (MW) value in 2019: BRL 998.00. No multicollinearity was observed in the multiple regression analysis.

## Data Availability

All data generated or analyzed during this study are included in this article. Further enquiries can be directed to the corresponding author.

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
