# Peer review of "Exploring Maternal Socio-Demographic Factors Shaping Children’s Dietary Patterns in Brazil: Results from the 2019 National Health Survey"

_ijerph, 2024, doi:10.3390/ijerph21080992_

Round 1

Reviewer 1 Report

Comments and Suggestions for Authors

The authors have derived the children's dietary patterns using principal component analysis and tested their association with maternal sociodemographic factors using linear regression models. This study used data from the Brazilian National Health Survey (PNS) conducted in 2019. The content of this manuscript appears appropriate for the Special Issue of “Women's Health, Pregnancy and Child Health” of this journal. However, the quality of the manuscript needs to be improved.

The following revision is recommended to further improve the quality of the manuscript.

Comment 1. ABSTRACT: There are some mistakes in the abstract:

a.       Line 1, please check whether it is appropriate to use the word “determine”.

b.       Line 24, please check with the quotation mark for the word “Healthy”.

Comment 2. INTRODUCTION: some key information needs to be added to this section for readers’ interests:

a.       Only standards, recommendations, and status in Brazil are mentioned. However, why the issue of child nutrition in Brazil is important to the world, and how is Brazil’s status compared to the world or other developing countries should be mentioned, as well as the WHO recommendations.

b.       Line 59-60, “weight gain” is not always bad, please think of another word to express precisely.

c.       Literature review is missing in the section. Please add information on research of the relationship between children’s diets and their mothers’ factors. Through the review, the authors could explain the added value of this research. The content of Line 65-66 is arbitrary.

Comment 3. METHOD:

It is not clear what is the dependent variable of the multiple linear regression. Is it the Z-score? Or is it a binary variable of “healthy” or “unhealthy” dietary pattern of a child? If it is the binary variable, then what is the cut-off of the z-score to generate the binary variable? It is better to answer all these questions and display a table after Table 3 to show how many children’ dietary patterns are grouped as “healthy” and “unhealthy”. If it is not clearly shown, readers might get confused and mixed up with the factor loadings of healthy and unhealthy dietary patterns in Table 3.

Comment 4. RESULTS:

Table 4 is missing, I can not find it in the manuscript. As far as I understand, the most important results of this study are shown in Table 4, however, it is missing.

Comment 5. DISCUSSION:

a.       Besides mothers’ socio-economic status, it might be more interesting to check the relationship of mothers’ physical activity and smoking behavior and children’s diets. The authors reported the descriptive analysis of the mothers’ physical activity and smoking behavior but did not report a regression analysis of them. Why is that?

b.       Line 284-285, the authors mentioned the drawback of using cross-sectional data, however, the authors could try the Instrumental Variable method to make their results more convincing.

Comments on the Quality of English Language

Minor editing of English language required

Author Response

Author's Reply to the Review Report (Reviewer 1)

The authors have derived the children's dietary patterns using principal component analysis and tested their association with maternal sociodemographic factors using linear regression models. This study used data from the Brazilian National Health Survey (PNS) conducted in 2019. The content of this manuscript appears appropriate for the Special Issue of “Women's Health, Pregnancy and Child Health” of this journal. However, the quality of the manuscript needs to be improved.

The following revision is recommended to further improve the quality of the manuscript.

Authors’ response: Thank you for your comments. We have revised the manuscript by addressing each comment point-by-point as described below.

Comments 1. ABSTRACT: There are some mistakes in the abstract:

Comments 1a. Line 1, please check whether it is appropriate to use the word “determine”.

Response 1a: Thank you for your suggestions for improvement. We have changed the verb determine to identify. (page 1, line 18 in the revised manuscript).

Comments 1b. Line 24, please check with the quotation mark for the word “Healthy”.

Response 1b: Thank you for your suggestions for improvement. We rephrase the text as follows: “The first consisted healthy and the second, unhealthy.” (page 1, lines 23-24 in the revised manuscript).

Comment 2. INTRODUCTION: some key information needs to be added to this section for readers’ interests:

Comments 2a. Only standards, recommendations, and status in Brazil are mentioned. However, why the issue of child nutrition in Brazil is important to the world, and how is Brazil’s status compared to the world or other developing countries should be mentioned, as well as the WHO recommendations.

Response 2a: Thank you for your comments and suggestions for improvement. We rephrase the text as follows: “National surveys indicate that ultra-processed foods account for at least half of the total energy intake in some high-income countries, such as the United Kingdom. In middle-income countries, these foods contribute between one-fifth and one-third of the total energy consumed [12-14]”. (page 2, lines 53-56 in the revised manuscript).

Comments 2b. Line 59-60, “weight gain” is not always bad, please think of another word to express precisely.

Response 2b: Thank you for your comments and suggestions for improvement. We rephrase the text as follows: “The environment that mothers create for their children can either promote healthy dietary patterns or lead to adverse effects on their nutrition, leading to unhealthy dietary behaviours that can increase the risk of excessive weight gain and chronic non-communicable diseases”. (page 2, lines 60-63 in the revised manuscript).

Comments 2c. Literature review is missing in the section. Please add information on research of the relationship between children’s diets and their mothers’ factors. Through the review, the authors could explain the added value of this research. The content of Line 65-66 is arbitrary.

 Response 2c: Thank you for your comments and suggestions. We have added references to the relationship between maternal factors and children's diet. We have also reworded the text, which was considered arbitrary, as follows: "However, a few studies have examined how specific Brazilian maternal socio-demographic factors influence children's dietary patterns [16-20]." (page 2, lines 69-71 in the revised manuscript).

Comment 3. METHOD: It is not clear what is the dependent variable of the multiple linear regression. Is it the Z-score? Or is it a binary variable of “healthy” or “unhealthy” dietary pattern of a child? If it is the binary variable, then what is the cut-off of the z-score to generate the binary variable? It is better to answer all these questions and display a table after Table 3 to show how many children’ dietary patterns are grouped as “healthy” and “unhealthy”. If it is not clearly shown, readers might get confused and mixed up with the factor loadings of healthy and unhealthy dietary patterns in Table 3.

 Response 3: Thank you for your suggestions for improvement. We rephrase the text as follows: “The dietary patterns were obtained by factor analysis extraction using principal component analysis (PCA). The Bartlett test and the Kaiser-Mayer-Olkin (KMO) coefficient were used to verify the applicability of the method [23]. Using Kaiser's classification for KMO values [24], the following ranges were applied to characterise the levels of acceptance: 0.00 to 0.49 as unacceptable, 0.50 to 0.59 as poor, 0.60 to 0.69 as fair, 0.70 to 0.79 as moderate, 0.80 to 0.89 as meritorious, and 0.90 to 1.00 as a perfect fit. For the Bartlett test, we assumed a type I error rate of 5%. To determine the number of factors, we considered (i) eigenvalues > 1 and (ii) the inflection point of the eigenvalues from the Cattell scree test (scree plot). Varimax orthogonal rotation was used to facilitate the interpretation of the factors. Factor loadings greater than 0.30 were considered to represent dietary patterns. The total variance explained by each factor was also considered to determine the number of factors to be retained. Cattell's scree plots and eigenvalues >1 were also considered for pattern selection. The orthogonal varimax transformation was used to facilitate the interpretation of the factor results. The naming of the identified patterns was based on interpretability and the characteristics of the food groups retained in each pattern. Factor scores were predicted for each child in the study. These scores were calculated by adding the standardised values in each pattern's z-score of the food groups. A higher score indicates greater adherence to the dietary pattern. Finally, associations between the children’s factor scores for each dietary pattern identified (dependent variables) and the maternal socio-demographic factors (independent variables) were examined using crude and multiple linear regression models with associations described using β-coefficients, 95% confidence intervals (95% CI), and p-values (p). Maternal socio-demographic factors that had a p < 0.20 in the crude regression were included in the multiple linear regression model using a backward stepwise procedure, retaining only those that were statistically significant (p < 0.05)”. (pages 3-4, lines 140-165 in the revised manuscript).

Comment 4. RESULTS: Table 4 is missing, I can not find it in the manuscript. As far as I understand, the most important results of this study are shown in Table 4, however, it is missing.

 Response 4: Thank you for your suggestions for improvement. We apologise for the absence of Table 4 in the first version of the manuscript. We have added the table in this revised version. (page 7-8, lines 223-230 in the revised manuscript).

Comment 5. DISCUSSION:

Comment 5a: Besides mothers’ socio-economic status, it might be more interesting to check the relationship of mothers’ physical activity and smoking behavior and children’s diets. The authors reported the descriptive analysis of the mothers’ physical activity and smoking behavior but did not report a regression analysis of them. Why is that?

 Response 5a: Thank you for your comments. We apologise for the absence of Table 4 in the first version of the manuscript. We have added it on this revised version. (Page, line in revised manuscript). We examined the relationship between mothers' physical activity and smoking behaviour and children's diet.. However, these variables were not statistically significant. See Table 4.

Comment 5b. Line 284-285, the authors mentioned the drawback of using cross-sectional data, however, the authors could try the Instrumental Variable method to make their results more convincing.

 Response 5b:

Thank you for your insightful suggestions. However, our study is primarily descriptive and does not aim to establish causal relationships between the variables observed. Consequently, we amended the text to state "inherently to the study design we looked at associations and not causality." (page 10, lines 312-314 in the revised manuscript).

We understand that the Instrumental Variables (IV) method is a robust approach for estimating causal relationships using observational data. This method is particularly valuable when standard regression estimates may be biased due to issues such as reverse causality, selection bias, measurement error, or unmeasured confounding effects. The key idea behind IV is to identify a third variable, known as the 'instrumental' variable, which is related to the variable of interest but not to the confounding factors, thus allowing for a more accurate estimation of the causal effect on the outcome measure.

Despite the potential benefits of the IV method, the primary focus of our current study is to describe associations rather than to confirm causality. Future research could indeed explore causal relationships using IV or other methodologies. Thank you again for your valuable feedback.

Reviewer 2 Report

Comments and Suggestions for Authors

Dear authors,

The paper is written very well, to the point and provides all the necessary details. 

I tried this link (https://www.ibge.gov.br/en/statis-163 tics/social/health/16840-national-survey-of-health.html?=&t=microdados) to access the data and was unable to. Please update the link.

I do have one concern and that is the description of the participants in the results section using colour of skin rather than just cultural heritage. Some readers may find this offensive and of little meaning. 

Lastly, include a subheading for limitations please. 

See attached PDF. 

Author Response

Author's Reply to the Review Report (Reviewer 2)

Dear authors,

The paper is written very well, to the point and provides all the necessary details. 

Authors’ response: Thank you for your comments. We have revised the manuscript by addressing each comment point-by-point as described below.

Comment 1: I tried this link (https://www.ibge.gov.br/en/statis-163 tics/social/health/16840-national-survey-of-health.html?=&t=microdados) to access the data and was unable to. Please update the link.

 Response 1: Thank you for your suggestions. We have updated the link: https://www.ibge.gov.br/estatisticas/sociais/saude/29540-2013-pesquisa-nacional-de-saude.html?edicao=9177&t=microdados (page 4, lines 173-175 in the revised manuscript).

Comment 2: I do have one concern and that is the description of the participants in the results section using colour of skin rather than just cultural heritage. Some readers may find this offensive and of little meaning. 

 Response 2: Thank you for your suggestions for improvement. We agree with this comment. We have changed to ethnicity in the revised manuscript.

Comment 3: Lastly, include a subheading for limitations please. 

 Response 3: Thank you for your suggestions. We have added a subheading for limitations. (page 10, line 311 in the revised manuscript).

Reviewer 3 Report

Comments and Suggestions for Authors

This is a good research study that contributes to the knowledge of infant feeding, however I have some concerns. 

Abstract

1. Can authors state the b-values and p-values for the regressions

2. Authors should conclude on the results in the abstract

3. Authors should do a better job in describing the methods in the abstract. 

Introduction

1. The meaning of line 40 isn't clear, because exclusive breastfeeding isn't the same as formula feeding 

2. Line 49- has some grammatical issues. 

3. State the hypothesis of this work in the introduction 

Material and methods

Questionnaire could be added as supplementary material

Many of the results discussed in the discussion isn't shown in the tables. Can authors show all relevant tables, because I believe there is room for more tables to be added. Table 4 is described but it isn't shown in the manuscript. 

Discussion

The authors should start this section with aims and hypothesis of the study before zooming into the results. 

The information on Line 225 and 231 should be link to results in this study

Information on line 245-254 should be linked to the results of this study

I am not sure information on line 307-314 is relevant to the write up. 

Thank you

Comments on the Quality of English Language

Few grammatical errors noted. Authors should check documents for grammatical errors. 

Author Response

Author's Reply to the Review Report (Reviewer 3)

This is a good research study that contributes to the knowledge of infant feeding, however I have some concerns. 

Authors’ response: Thank you for your comments. We have revised the manuscript by addressing each comment point-by-point as described below.

Comments 1. Abstract

Comments 1a. Can authors state the b-values and p-values for the regressions

Response 1a: Thank you for your comments and suggestions. However, we have not been able to incorporate them. According to the IJERPH rules, the abstract should not exceed 200 words.

Comments 1b. Authors should conclude on the results in the abstract

Response 1b: Thank you for your comments and suggestions for improvement. We have revised the manuscript to clarify as follows. “We conclude that socio-demographic factors may influence the quality of the food offered to children. Nevertheless, advocating for public policies promoting nutritious complementary diets emphasising natural and minimally processed foods remains crucial for children whose mothers do not possess these favourable socio-demographic characteristics.” (page 1, lines 29-33 in the revised manuscript).

Comments 1c. Authors should do a better job in describing the methods in the abstract. 

Response 1c: Thank you for your comments and suggestions for improvement. We have revised the manuscript to clarify as follows. “Data from the 2019 Brazilian National Health Survey were used in this cross-sectional study. Mothers of 1,616 children aged 6-23 months reported on their children's dietary intake. Dietary patterns were identified using principal component analysis, and their associations with maternal sociodemographic characteristics were assessed using linear regression models”. (page 1, lines 19-23 in the revised manuscript).

Comments 2. Introduction

Comments 2a. The meaning of line 40 isn't clear, because exclusive breastfeeding isn't the same as formula feeding 

Response 2a: Thank you for your suggestions for improvement. We rephrase the text as follows: “The first two years of a child's life are a critical period for optimal growth, health, and behavioural development [1]. Complementary feeding is recommended to begin at six months and continue until 23 months, although breastfeeding may continue beyond this period. Complementary feeding provides essential nutrients when breast milk or formula milk alone is insufficient to meet nutritional requirements [2,3]”. (page 1, lines 39-42 in the revised manuscript).

Comments 2b. Line 49- has some grammatical issues. 

Response 2b: Thank you for your suggestions for improvement. We rephrase the text as follows: “The consumption of fresh or minimally processed foods and culinary ingredients has decreased, while the consumption of ultra-processed foods (UPF) has increased [8-10]” (page 2, line 49 in the revised manuscript).

Comments 2c. State the hypothesis of this work in the introduction 

Response 2c: Thank you for your suggestions for improvement. We state the hypothesis of this work in the text as follows: “We believe that these maternal characteristics influence the quality of the food that is offered to children". (page 2, lines 66-67 in the revised manuscript).

Comments 3. Material and methods

Comments 3a. Questionnaire could be added as supplementary material

Response 3a: Thank you for your suggestions. The questionnaire has been added as Supplementary Material.

Comments 3b. Many of the results discussed in the discussion isn't shown in the tables. Can authors show all relevant tables, because I believe there is room for more tables to be added. Table 4 is described but it isn't shown in the manuscript. 

Response 3b: Thank you for your suggestions for improvement. We agree with this comment. We apologise for the absence of Table 4 in the first version of the manuscript. We have added the table 4 in this revised version.

Discussion

Comments 4a. The authors should start this section with aims and hypothesis of the study before zooming into the results. 

The present study identified two dietary patterns: healthy and unhealthy. The healthy pattern explained the highest proportion of total variance and best represented the dietary intake of Brazilian children under two years of age. This pattern included unprocessed or minimally processed food groups related to fruits, beans or legumes, vegetables, meat or eggs, and non-breast milk.

Comments 4b. The information on Line 225 and 231 should be link to results in this study

Response 4b: Thank you for your comments and suggestions for improvement. We have linked to results in this study as follows. “Although most children in the present study followed the healthy pattern, 74.2% of children consumed at least one UPF on the previous day. In line with these findings, data from the first Brazilian National Survey on Child Nutrition (ENANI-2019) revealed a concerning prevalence of UPF consumption among children aged 6-23 months, reaching 80.5% [27]. Cainelli et al. found that 79.4% of children aged 1 to 2 years consumed some type of UPF and that consumption of such foods was associated with household socio-economic and demographic factors [28].”. (page 9, lines 248-255 in the revised manuscript).

Comments 4c. Information on line 245-254 should be linked to the results of this study

Response 4c: Thank you for your valuable feedback. We have linked to results in this study as follows. “These findings are consistent with both international and national studies that have reported similar associations [6,30,37-40]. For example, Campos et al. highlighted that mothers can positively or negatively influence their children's eating habits and diet quality [41]. In another study, Paramashanti et al. highlight that mothers or caregivers with higher levels of education are likely to be more knowledgeable about child nutrition and health literacy and to better understand the messages conveyed about child nutrition [42]. Similarly, a systematic review by Gutiérrez-Camacho et al., provides evidence of a positive association between maternal education level, higher household income, higher maternal age, and healthy dietary patterns, and a negative association between these factors and unhealthy dietary patterns [43]”. (page 9, lines 269-278 in the revised manuscript).

Comments 4d. I am not sure information on line 307-314 is relevant to the write up. 

Response 4d: Thank you for your suggestion. We agree with this comment and have removed the text as follows: "In Brazil, the Ministry of Health has strengthened the Dietary Guidelines for Bra-zilian Children under 2 Years of Age, advocating against offering ultra-processed foods during this crucial phase. This recommendation was grounded in comprehensive evidence highlighting the health and environmental implications linked to their consumption [3,8,26]".

Round 2

Reviewer 1 Report

Comments and Suggestions for Authors

The quality of the manuscript has been well improved after revisions addressing my comments and advice from the previous review. I don't have any further comments. Good luck to the authors. 

Author Response

Author's Reply to the Review Report (Reviewer 1) – Round 2

Comment 1: The quality of the manuscript has been well improved after revisions addressing my comments and advice from the previous review. I don't have any further comments. Good luck to the authors. 

Authors’ response: Thank you for taking the time to review our manuscript. We appreciate your valuable comments and suggestions, which have undoubtedly strengthened our manuscript.